# A new classification of satellite derived liquid water cloud regimes at cloud scale

Claudia Unglaub[1], Karoline Block[1], Johannes Mülmenstädt[1,2], Odran Sourdeval[1,3], and Johannes Quaas[1]

[1]Universität Leipzig, Institute for Meteorology, Stephanstr. 3, 04103 Leipzig
[2]now at Pacific Northwest National Laboratory, Richland
[3]now at Laboratoire d'Opique Atmosphérique, Université de Lille

**Correspondence:** Johannes Quaas (johannes.quaas@uni-leipzig.de)

**Abstract.** Clouds are highly variable in time and space affecting climate sensitivity and climate change. To study and distinguish the different influences of clouds on the climate system it is useful to separate clouds into individual cloud regimes. In this work we present a new cloud classification for liquid water clouds at cloud scale defined using cloud parameters retrieved from combined satellite measurements from CloudSat and CALIPSO. The idea is that cloud heterogeneity is a measure that allows to distinguish cumuliform and stratiform clouds, and cloud base height a measure to distinguish cloud altitude. The approach makes use of a newly-developed cloud-base height retrieval. Using three cloud base height intervals and two intervals of cloud top variability as an inhomogeneity parameter provides six new liquid cloud classes. The results show a smooth transition between marine and continental clouds as well as between stratiform and cumuliform clouds in different latitudes at the high spatial resolution of about $20\,\mathrm{km}$. Analyzing the micro- and macrophysical cloud parameters from collocated combined MODIS, CloudSat and CALIPSO retrievals shows distinct characteristics for each cloud regimes that are in agreement with expectation and literature. This demonstrates the usefulness of the classification.

## 1 Introduction

Clouds affect the climate system in a wide varieties of ways. They influence outgoing solar and terrestrial radiation and therefore the Earth's temperature, produce precipitation, transport heat and moisture and interact with the surrounding atmosphere including aerosols on different time and spatial scales. They exhibit a high variability from minutes to days in time and meters to thousands of kilometer in space. Because of their complexity, the response of clouds to perturbations remains one of the largest uncertainties in climate prediction (e. g., Boucher et al., 2013). Different cloud regimes have different impacts on climate. Low clouds and optical thick clouds contribute to cooling the climate system because their high albedo effect dominates their effect on emitted longwave radiation back to space (Hartmann et al., 1992) whereas thin medium and high altitude clouds rather contribute to warming the climate system (Dhuria and Kyle, 1990).

Consequently, since the early start of meteorological research, clouds have been classified (Howard, 1803). A fundamental distinction usually is made by cloud altitude (often in three classes of low, middle and high tropospheric clouds, WMO, 1975) as well as the separation of stratiform and cumuliform clouds (WMO, 1975, 2017).

Cloud types are often defined using the dynamical state of the atmosphere, or, alternatively, using cloud parameters themselves, or a mix of both. Dynamical regimes are often based on large scale, mid-tropospheric vertical velocity ($\omega_{500\,\mathrm{hPa}}$) derived from meteorological model reanalysis (e. g., Bony et al., 2004; Norris and Weaver, 2001). Also the lower-tropospheric stability (LTS; Klein and Hartmann, 1993) or, alternatively, the estimated inversion strength (EIS; Wood and Bretherton, 2006) or estimated low-level cloud fraction (Park and Shin, 2019; Shin and Park, 2019) have been used to characterise low-level

clouds; some studies have used a combination of mid-tropospheric vertical velocity and LTS/EIS (Su et al., 2010; Medeiros and Stevens, 2011). Tselioudis et al. (2000) use the sea level pressure to define three different dynamical cloud types in the northern midlatitudes, and Ringer and Allan (2004) combine sea surface temperature and $\omega_{500\,\mathrm{hPa}}$. As a prime example of the other method, the International Satellite Cloud Climatology Project (ISCCP) cloud classification uses cloud optical thickness, $\tau_{\mathrm{c}}$, and cloud top pressure $p_{\mathrm{top}}$ to separate 49 or, in a simplified version, nine cloud types (Rossow and Schiffer, 1999). By

applying a clustering algorithm to these ISCCP cloud classes, Jakob et al. (2005) defined four cloud regimes in the tropical western Pacific using $\tau_{\mathrm{c}}$, $p_{\mathrm{top}}$ and the total cloud cover $f_{\mathrm{tot}}$. Extending and simplifying this approach for climate model evaluation, Williams and Webb (2008) selected different cloud regimes in particular geographical regions using cloud albedo, $p_{\mathrm{top}}$ and total cloud cover, $f_{\mathrm{tot}}$. Such a regime definition was also found useful in the context of the analysis of aerosol optical depth-cloud droplet concentration using satellite data in the study of Gryspeerdt and Stier (2012).

We are interested to statistically analyse aerosol-cloud interactions in satellite data beyond the aerosol-droplet concentration relationship. In order to identify aerosol-cloud interactions, Stevens and Feingold (2009) suggested that it is necessary to do so for individual cloud regimes (see also Mülmenstädt and Feingold, 2018). However, a dynamical regime definition is hampered by the problem of a rather coarse resolution of the reanalysis data (50-100 km currently) and the problem that thus such cloud regimes are not able to separate clouds at the scale of individual cloud regimes (Nam and Quaas, 2013). In turn, the approach

to use the ISCCP cloud definition (e.g., Jakob et al., 2005) is not useful to analyse aerosol-cloud interactions if one is interested in analysing how cloud fraction and cloud albedo co-vary with the aerosol since these quantities are fixed by the clustering method.

In this work we present a new cloud classification at cloud scale using the cloud base height indicating meteorological conditions and separating cloud altitude, and the cloud top variability as an inhomogeneity parameter separating between stratiform

and cumuliform clouds. The collocated satellite data and the high spatial resolution defined as the Clouds and the Earth's Radiant Energy System (CERES) footprint size of about $20\,\mathrm{km}$ allow a cloud class based analysis of cloud parameter reflecting the high spatial and temporal variability.

## 2  Satellite data

Our studies rely on retrievals of two active satellite instruments, the Cloud Profiling Radar (CPR; Stephens et al., 2008; Haynes

et al., 2009) onboard CloudSat and the Cloud-Aerosol LIdar with Orthogonal Polarization (CALIOP; Winker et al., 2007, 2009) onboard Cloud-Aerosol Lidar and Infrared Pathfinder Satellite Observations (CALIPSO), as well as the passive Moderate Resolution Imaging Spectroradiometer (MODIS; Barnes et al., 1998; King et al., 2003; Platnick et al., 2003) instrument

onboard Aqua. These satellites are part of the A-Train satellite constellation (Stephens et al., 2002, 2018), a group of satellites flying along nearly the same polar orbital track crossing the equator at about 13:30 local time and providing a global data coverage between 82°N and 82°S (Winker et al., 2007; Tanelli et al., 2008). The sun-synchronous polar orbit repeats the same ground track every 16 days retaining its size and shape (Stephens et al., 2008).

The CALIPSO CloudSat CERES and MODIS merged product (CCCM dataset) contains collocated data from CALIOP, CPR, MODIS and the broadband radiometer CERES providing comprehensive informations about clouds, aerosols and radiation fluxes in high vertical and horizontal resolution (Kato et al., 2010, 2011), merged to the CERES footprint of about 20 km horizontal size. It is this combined product that is the basis for our analysis. The collocation of these various retrievals with different spatial resolution requires a two step process. In the first step the vertical cloud profiles as provided in the Vertical feature mask from CALIPSO (Winker et al., 2007) and the 2B-Cldclass product from CloudSat (Stephens et al., 2008) are collocated on a horizontal 1 km × 1 km grid. Each grid point contains three vertical cloud profiles from CALIPLSO, and one from CloudSat, these are used to derive cloud top heights and cloud base heights (Kato et al., 2010). With this merging procedure about 85% of the cloud top heights and 77% of the cloud base heights are derived from CALIPSO measurements. The second step starts with collocating horizontally the merged vertical cloud profiles with CERES footprints of about 20 km size by selecting the CERES footprints with maximum overlap with the CALIPSO-CloudSat ground track. Because the horizontal resolution of CERES is much coarser than the horizontal resolution of the combined CloudSat/CALIPSO vertical cloud profiles, at each grid box, CloudSat/CALIPSO clouds groups are defined to retain the statistical cloud geometric information.

The temperature profiles included in the CCCM dataset are derived at computational levels from the CERES Meteorological, Ozone, and Aerosol (MOA) analysis. They come from the Global Modeling and Assimilation Office (GMAO) Goddard Earth Observing System (GEOS)-4 (Bloom et al., 2005) Data Assimilation System reanalysis before November 2007 and GEOS-5 (Rienecker et al., 2008) thereafter (Kato et al., 2014) with a temporal resolution of 6 h and a spatial resolution of $1° \times 1°$ (Kato et al., 2011).

A key parameter used in this paper is cloud base height, $H_{\mathrm{base}}$. This relies on a new retrieval on the basis of CALIPSO lidar described by Mülmenstädt et al. (2018) that assumes that $H_{\mathrm{base}}$ is constant in a scene, and that the lowest lidar return within columns that do not fully attenuate the lidar beam is representative for $H_{\mathrm{base}}$. Besides $H_{\mathrm{base}}$, also cloud top height, $H_{\mathrm{top}}$, is used as derived from the CALIPSO in the merged vertical cloud profiles. Mülmenstädt et al. (2018) thoroughly examined the cloud-base altitude using ground-based ceilometer data as reference. The root-mean-square-error on retrieved cloud base height was in the range of 400 to 700 m, and biases much lower at 5 to 50 m. Both parameters are defined here with respect to the surface altitude.

Cloud top temperature $T_{\mathrm{top}}$ is taken from MODIS and CloudSat/CALIPSO as derived from their respective $H_{\mathrm{top}}$ assigned to $T_{\mathrm{top}}$ using the temperature profile. Further, cloud optical thickness, $\tau_{\mathrm{c}}$, and, cloud droplet effective radius, $r_{\mathrm{eff}}$, as derived from MODIS measurements are analysed. We use retrievals that apply the 3.7 μm channel (Platnick, 2002; Platnick et al., 2003; Painemal and Zuidema, 2011).

Daytime data are used, and high latitudes (polewards of 60°) are excluded to avoid biases in the retrieved cloud optical prop-

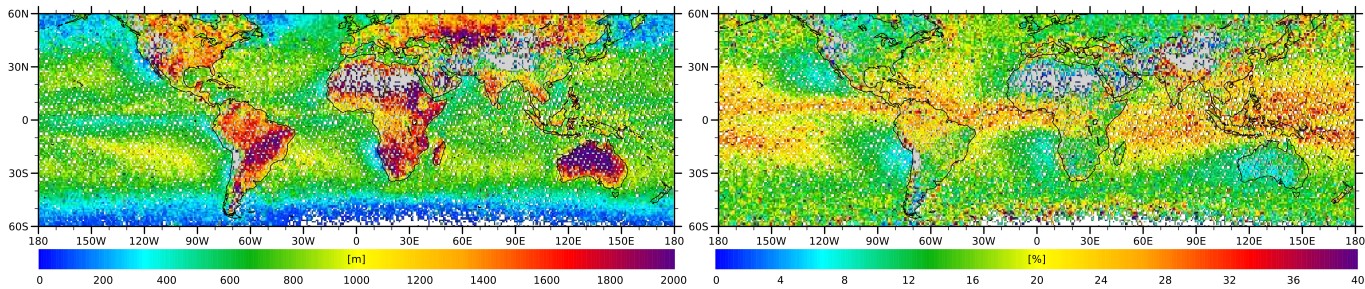

**Figure 1.** Time-average distributions of daily values for the 2007 - 2010 period of CALIPSO retrievals as reported in the CCCM dataset for liquid-water clouds for (a) cloud-base altitude (using the retrieval method of Mülmenstädt et al., 2018), and (b) cloud top height variability.

erties from MODIS (Zeng et al., 2012; Grosvenor et al., 2018). Our studies investigate only liquid water clouds. These are defined as clouds where $T_{\text{top}}$ derived from both MODIS and CloudSat/CALIPSO are larger than $273\,\text{K}$.

## 3 Definition of the cloud classes

The liquid cloud classes are defined at the scale of the CERES footprint size of about $20\,\text{km}$ as horizontal resolution, at which the CCCM dataset is generated. This results in one liquid cloud type per footprint after the cloud classification process described in the following.

### 3.1 Cloud base height

The first cloud parameter selected to define the cloud classes is cloud base height over ground $H_{\text{base}}$. This is consistent with
100 the WMO definition of cloud altitude (WMO, 1975, 2017). $H_{\text{base}}$ is used from the retrieval approach of Mülmenstädt et al. (2018), applied to the CCCM dataset. $H_{\text{base}}$ of a multilayer clouds is defined as the lowest $H_{\text{base}}$ in this cloud group. In Fig. 1 the global distribution of the averaged $H_{\text{base}}$ of the four completely available years of CCCM data from 2007 to 2010 is shown. One can see a clear contrast between land and ocean and between higher and lower latitudes. The lowest $H_{\text{base}}$ are located over the ocean in the storm track regions in mid latitudes whereas the highest $H_{\text{base}}$ can be found over land for example over
105 the Amazon rain forest or Australia.

To separate different cloud base height classes the probability density function (PDF) of the global spatiotemporal distribution of $H_{\text{base}}$ shown in Fig. 2 is used. Three cloud base height classes are selected which are the round numbers that approximately correspond to the terciles of the distribution, which is the median $\pm 300\,\text{m}$. With this definition, *low* clouds are defined as those with $H_{\text{base}} \leq 350\,\text{m}$; *middle* clouds for $350\,\text{m} < H_{\text{base}} \leq 950\,\text{m}$; and *high* (liquid) clouds for $H_{\text{base}} > 950\,\text{m}$.

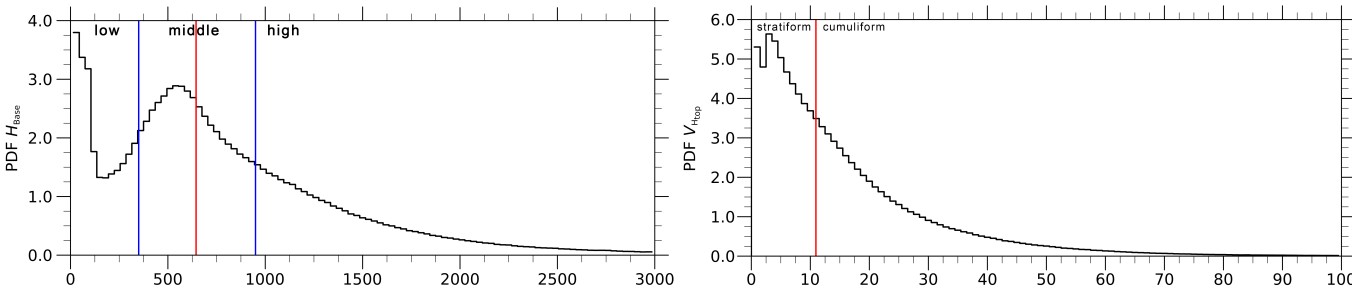

**Figure 2.** PDFs of the spatiotemporal distributions of (a) of cloud base height (m) and (b) cloud-top height variability (%), for daily data for the four-year period 2007 to 2010. The red lines indicate the median of the PDFs and the two blue lines in (a) represent the borders of the cloud base classes.

## 3.2 Cloud top variability as inhomogeneity parameter

Only using $H_{\mathrm{base}}$ one cannot distinguish between cumuliform and stratiform clouds as proposed by the WMO. We propose to use an inhomogeneity parameter and define stratiform clouds as homogeneous, cumuliform clouds as inhomogeneous clouds. Cloud optical thickness, $\tau_{\mathrm{c}}$, is often used to describe the inhomogeneity of a cloud or cloud field and to separate clouds into homogeneous and inhomogeneous clouds. The ISCCP cloud classification uses $\tau_{\mathrm{c}}$ itself to classify stratiform and cumuliform

clouds, defining clouds with high $\tau_{\mathrm{c}}$ as stratiform (Rossow and Schiffer, 1999). In a more advanced approach, the horizontal variablity of $\tau_{\mathrm{c}}$ derived from MODIS measurements is defined as cloud inhomogeneity parameter to distinguish stratiform and cumuliform clouds (Oreopoulos and Cahalan, 2005).

With the definition proposed here for cloud regimes, however, we aim to analyse adjustments to aersol-cloud interactions (e.g., Mülmenstädt and Feingold, 2018), i.e. the response of cloud liquid water path to perturbations in cloud droplet concen-

120 trations (e.g., Gryspeerdt et al., 2019). It is thus impossible to use $\tau_{\mathrm{c}}$ to define cloud regimes, since this would constrain liquid water path.

We thus propose to define cloud inhomogeneity based on the cloud top height variability. Cloud top height is related to $\tau_{\mathrm{c}}$ and also $H_{\mathrm{base}}$, but its variability is independent of it. The idea is that clouds with horizontally homogeneous top heights are more stratiform, and those with horizontally inhomogeneous top heights more cumuliform. Cloud top height variability is defined

here as the average relative deviation of cloud top heights from its footprint mean. Preliminary analysis of the cloud-top height variability at the scale of a CERES footprint, given the MODIS resolution, often is not well defined, at least in broken-cloud situations. This is due to the too low number of MODIS retrievals within a CERES footprint. Thus, the variability in the two adjacent footprints (adjacent along the A-Train ground track), in addition to the footprint at nadir below the satellite is used, and the average cloud-top height variability weighted by cloud occurrence in the three footprints, is used.

In Fig. 1, the global distribution of the mean cloud top variability from 2007 to 2010 is shown. No clear land-ocean contrast is seen, but the distribution is characterized by a latitudinal gradient with the highest values of cloud top variability in the tropics in the shallow cumulus regions and along the Intertropical Convergence Zone (ITCZ). In the shallow cumulus regions

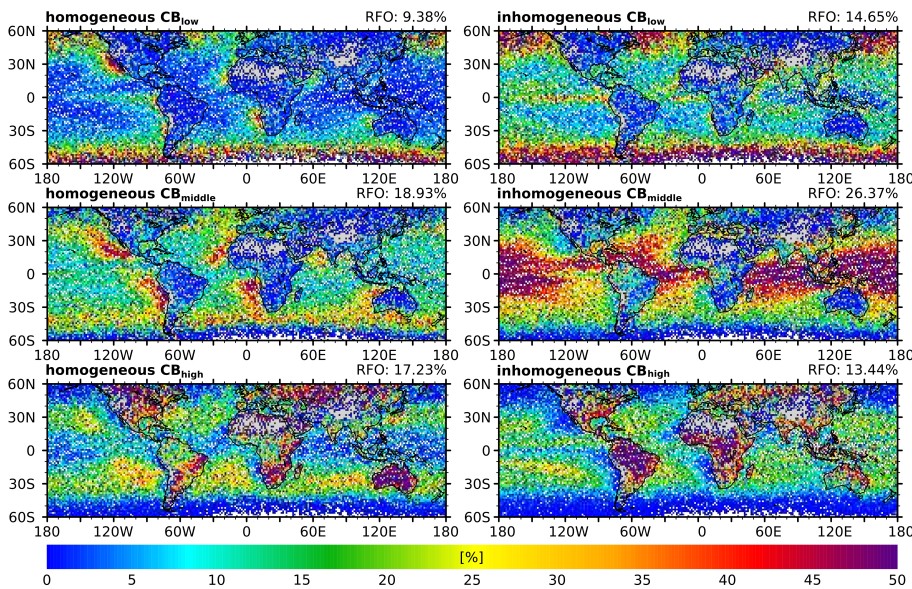

**Figure 3.** Relative frequency of occurrence of the six cloud classes separated using the three classes of cloud-base height and two classes of cloud-top height heterogeneity. Top row - low, middle - mid, bottom - high clouds; left column - homogeneous (stratiform), right column - heterogeneous (cumulisform) clouds.

towards the western parts of the oceans in the sub-tropics, the variability is – with values between 20 and 30% – about as large as in the Tropics for the Indian and Pacific oceans, it is, however, somewhat lower in particular in the southern Atlantic ocean.

At mid- to high latitudes the mean cloud top variability decreases in general, compared to the tropical regions. Although at low latitudes, the stratocumulus decks west South Africa, South America and North America show the smallest mean cloud top variabilities. These features are consistent with the hypothesis that the heterogeneity metric is useful to distinguish stratiform and cumuliform clouds. However, it is not a perfect classification into stratiform and cumuliform clouds e.g. for suppressed shallow convection where the cloud top altitude is dictated by subsidence. The PDF of the cloud top variability shown in Fig. 2

is used to make this distinction. The median at about 11% separates the more stratiform (homogeneous) clouds from more cumuliform (inhomogeneous) clouds creating two inhomogeneity cloud classes.

### 3.3    Geographical distribution of the cloud regimes

The three cloud base classes and two inhomogeneity cloud classes are now combined to define six new liquid water cloud types or cloud regimes. The global distribution of relative frequencies of occurence (RFOs) of these cloud classes is shown in Fig. 3.

Clouds with low and mid base altitude tend to be more heterogeneous, and clouds with high base altitude, more homogeneous. Low and mid clouds tend to occcur over ocean rather than land, although the contrast is less strong for stratiform clouds.

Most of the liquid water clouds with low $H_{\mathrm{base}}$ are located over the ocean in the storm track regions in mid to high latitudes. Only a small amount of the low clouds occur in low latitudes. Homogeneous low clouds concentrate in mid latitudes especially in the Southern hemisphere and in narrow coastal stripes west of North and South America and North and South Africa indicating parts of the typical stratocumulus clouds in these regions with very low $H_{\mathrm{base}}$. The occurrence is highest over regions with relatively low sea surface temperatures. The inhomogeneous clouds in this cloud base class occur mainly in the mid latitudes in both hemispheres though a small amount can be found in tropical regions especially along the ITCZ in the east Pacific.

Almost all mid-level-base clouds are marine clouds located over the oceans in low latitudes. Especially in the tropics along the ITCZ in the Indian ocean and in the west Pacific this cloud class is frequent. Inhomogeneous clouds in this cloud base class extend in low latitudes around the entire globe leaving out the stratocumulus decks and concentrate mainly in shallow cumulus regions and along the ITCZ.

In contrast to the cloud base classes of lower $H_{\mathrm{base}}$ most of the high clouds occur over land. However, a non-negligible amount can be found over the ocean in low latitudes. Only in higher latitudes over the ocean and in the stratocumulus regions in the east Pacific and east Atlantic almost no clouds with $H_{\mathrm{base}} > 950\,\mathrm{m}$ are found. A significant amount of homogeneous clouds in this cloud base class are located over land with maxima over South Africa, Australia and north west Asia. Over the ocean they cover two bands in both hemispheres at around $30°$ leaving out roughly the areas covered by the inhomogeneous mid-level-base clouds. The inhomogeneous high-base clouds occur mainly over land with maxima over rain forest regions in South America and middle Africa. Over ocean these clouds can be found equally distributed to inhomogeneous clouds in low latitudes except in the stratocumulus decks.

## 4 Cloud properties in the six cloud regimes

The key reason to define cloud regimes is that clouds are supposed to show different characteristics in these regimes. The hypothesis is that their response to perturbations e.g. of aerosol concentrations possibly can be identified more clearly in analyses when focusing on individual regimes (Mülmenstädt and Feingold, 2018). The goal of this section is to demonstrate the usefulness of the separation in cloud regimes according to the six classes defined in the previous section. To this end, the two main bulk cloud quantities are investigated, namely the cloud liquid water path, $L$, and the cloud droplet number concentration, $N_{\mathrm{d}}$. Both are computed on the basis of the MODIS bi-spectral retrievals as reported in the CCCM dataset. $N_{\mathrm{d}}$ is computed from the retrieved cloud optical thickness and cloud-top droplet effective radius following Grosvenor et al. (2018) and the parameters defined in Quaas et al. (2006).

Cloud droplet number concentration is a key quantity when assessing aerosol-cloud interactions and cloud radiative effects (e.g., Grosvenor et al., 2018). It depends on chemical composition and size distribution of the precursor aerosol, as well as cloud-base vertical velocity (Barahona et al., 2011). It is also very much influenced by cloud- and precipitation microphysical processes (Wood et al., 2012) as well as cloud-top and cloud-side entrainment.

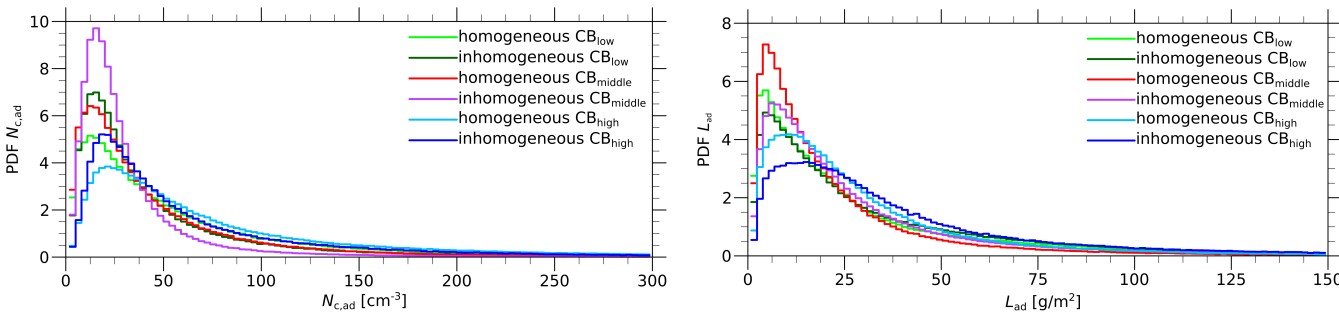

**Figure 4.** PDFs of the spatio-temporal distribution of four years (2007 – 2010) of MODIS retrievals as reported in the CCCM dataset at the $20 \times 20\,\text{km}^2$ horizontal resolution at nadir below the A-Train satellite constellation, between $60°\,\text{S}$ and $60°\,\text{N}$. Liquid clouds are selected. Light green - low, homogeneous clouds; dark green - low, heterogeneous clouds; red - middle, homogeneous clouds; purple - middle, heterogeneous clouds; light blue - high, homogeneous clouds; dark blue - high, heteorgeneous clouds. (a) for droplet number concentration, $N_\text{d}$ ($\text{cm}^{-3}$), (b) liquid water path, $L$ ($\text{g}\,\text{m}^{-2}$).

| Cloud Classes | Low | | Middle | | High | |
|---|---|---|---|---|---|---|
| | $N_\text{d}$ | $L$ | $N_\text{d}$ | $L$ | $N_\text{d}$ | $L$ |
| Homogeneous, mean | $76\,\text{cm}^{-3}$ | $33\,\text{g}\,\text{m}^{-2}$ | $49\,\text{cm}^{-3}$ | $24\,\text{g}\,\text{m}^{-2}$ | $91\,\text{cm}^{-3}$ | $33\,\text{g}\,\text{m}^{-2}$ |
| median | $39\,\text{cm}^{-3}$ | $19\,\text{g}\,\text{m}^{-2}$ | $29\,\text{cm}^{-3}$ | $14\,\text{g}\,\text{m}^{-2}$ | $54\,\text{cm}^{-3}$ | $21\,\text{g}\,\text{m}^{-2}$ |
| Inhomogeneous, mean | $55\,\text{cm}^{-3}$ | $36\,\text{g}\,\text{m}^{-2}$ | $34\,\text{cm}^{-3}$ | $32\,\text{g}\,\text{m}^{-2}$ | $73\,\text{cm}^{-3}$ | $45\,\text{g}\,\text{m}^{-2}$ |
| median | $28\,\text{cm}^{-3}$ | $21\,\text{g}\,\text{m}^{-2}$ | $21\,\text{cm}^{-3}$ | $19\,\text{g}\,\text{m}^{-2}$ | $42\,\text{cm}^{-3}$ | $28\,\text{g}\,\text{m}^{-2}$ |

**Table 1.** Average (blue) and median (red) values of $N_\text{d}$ and $L$ for the six particular cloud classes derived from their PDFs (Fig. 4).

Fig. 4 shows the global PDF of $N_\text{d}$ for the six cloud regimes, and Table 1 summarizes the mean and median values. The values are clearly distinct between the six classes. One key feature is that for all cloud-base heights, the homogeneous clouds contain fewer droplets than the heterogeneous ones. This is consistent with the expectation that heterogenous, convective clouds are driven by stronger updraughts. In terms of the altitude classes, low-base clouds show more droplets than mid-level-base clouds. This is a feature of the geographical distribution: both types occur mostly over oceans, but the mid-level-base clouds are more prevalent over the pristine parts of the oceans. The highest $N_\text{d}$ is observed for the high clouds, due to the fact that high clouds mostly occur over continents.

Also in terms of liquid water path, $L$, the clouds in the six classes are distinct. Homogeneous, i.e. more stratiform clouds, are thinner than the heterogeneous counterparts in each altitude class. This is consistent with the fact that convective clouds tend to develop more in the vertical, compared to stratiform clouds. Among the cloud altitude classes, $L$ is smallest for mid-level-base clouds and largest for high clouds. Note that these clouds are only the liquid-water clouds, so that the vertical development is limited by the $0°\text{C}$ level in our definition. Here, low clouds have the largest potential to develop in the vertical and yet remain

liquid. Over land, where high cloud bases are prevalent, the 0° level is reached at higher altitudes, allowing these clouds to develop further in the vertical. Due to the choice made here to investigate only liquid water clouds, the behaviour is different in different latitudes and seasons, due to the fact that the freezing level is at lower heights in higher latitudes and winter times. More detail of the geographical variation of the cloud regimes is provided in the Appendix, where the cloud properties for the different regimes are compared for land vs. ocean, and Tropics vs. Extratropics.

## 5 Summary and conclusions

The goal of the present study was to overcome limitations in the definition of cloud regimes. Such a definition is desirable e.g. in the context of studying aerosol-cloud interactions. Previous approaches were either at the comparatively very coarse resolution of meteorological re-analyses or used cloud parameters that are, however, the ones to study in aerosol-cloud interaction and thus cannot be used to stratify the data. Also, previous approaches were not very compatible with the standard WMO definitions. Here, we propose six cloud regimes for liquid clouds, separated by (i) cloud-base height and (ii) cloud top-height variability as an inhomogeneity parameter. Both parameters are derived from active remote sensing satellite measurements and are thus available at the scale of satellite retrievals. They are evaluated using a four year (2007 to 2010) dataset of combined A-Train satellite data in the CCCM dataset. A new approach to retrieve cloud-base altitude from spaceborne lidar has recently been developed and applied here. The geographical distributions of the frequency of occurrence of the six cloud regimes shows desirable features: oceanic and continental clouds are smoothly separated, and typical cloud regimes such as stratocumulus decks are readily identified. In order to demonstrate the usefulness of the cloud regimes, cloud parameters not used to define the regimes, but useful to study e.g. aerosol-cloud interactions, have been analysed. The selected parameters are cloud droplet concentration and cloud liquid water path. From the analysis it is evident that the cloud regimes show different characteristics in both quantities, i.e. the cloud types are clearly distinct. In particular, expected features of homogeneous (interpreted as stratiform) and heterogeneous (interpreted as cumuliform) clouds appear, as to features related to predominant aerosol sources and boundary-layer dynamics.

In future work, the study could be enhanced to study all clouds, and not just the liquid-water ones as done in the present study. While the cloud classification method could be adapted in a straightforward way, this would require a new analysis of how the classes differ in their characteristics. The current study is limited by the fact that it can only be applied to the ground-track below the A-Train lidar and radar retrievals. However, approaches exist to infer cloud-base altitude also from passive, multi-angle measurements (Böhm et al., 2019). An adaptation of our method to these swath data would allow to analyse much larger data volumes.

## Appendix A: Regime analysis by large-scale region

More detail about the characterization of $N_{\mathrm{d}}$ and $L$ by cloud regime is provided in Figs. A1 and A2 and summarized in Table A1. The PDFs of $N_{\mathrm{d}}$ (Fig. A1) and $L$ (Fig. A2) are separated into oceanic and continental surfaces, and between Tropics

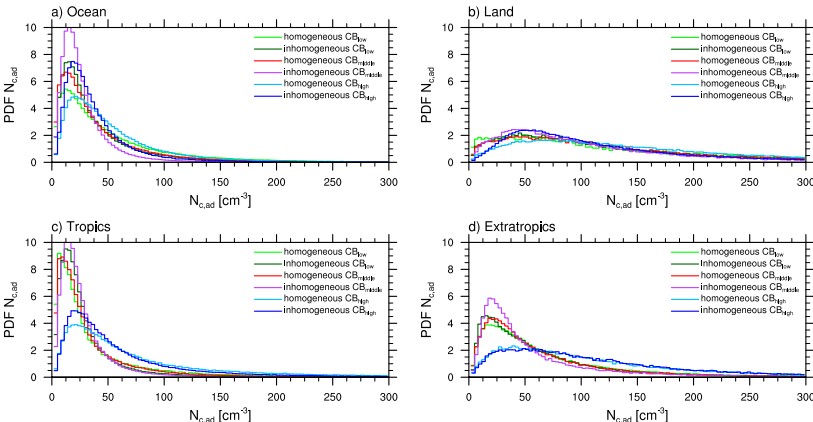

**Figure A1.** As Fig. 4a, but separately for (a) ocean, (b) land, (c) Tropics (20°S- 20°N) and (d) Extratropics (40°S - 60°S and 40°N - 60°N).

and Extratropics, respectively. The droplet concentrations are somewhat lower over ocean compared to the global mean, but are by factors of two to four higher over land (there are many more data points for liquid-water cloud retrievals over ocean than over land) in all cloud regimes, consistent with the expectation. The result of smaller $N_\mathrm{d}$ for inhomogeneous vs. homogeneous clouds holds for all categories over both, land and ocean. Liquid water path on average is slightly lower over ocean, slightly larger over land, except for the high clouds where things are rather similar. That inhomogeneous clouds have higher $L$ holds true over both land and ocean, with the exception of the low clouds over land. Clouds in the Tropics have larger $N_\mathrm{d}$ than in the Extratropics, and they also have larger $L$ (except for those with high cloud bases).

*Data availability.* All analyses are based on the publicly available CCCM dataset (Kato et al., 2010, 2011).

*Author contributions.* CU and JQ designed the research with input from all authors. OS and JM prepared the satellite data. CU and KB performed the data analysis with support by all other authors. CU and JQ wrote the manuscript with input from all authors.

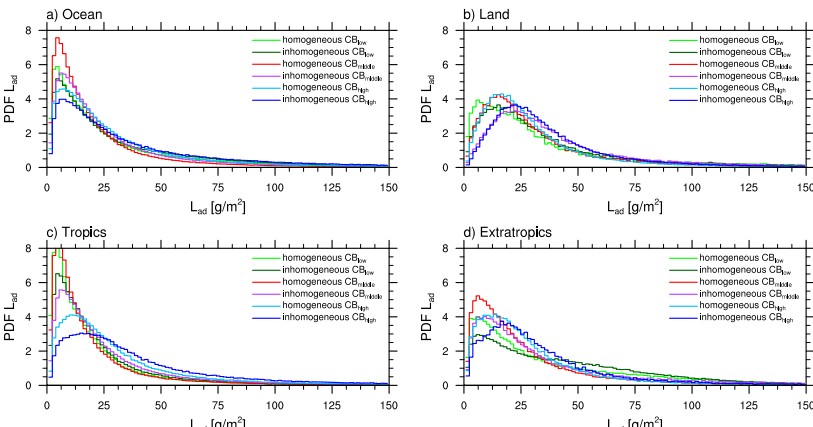

**Figure A2.** As Fig. 4b, but separately for (a) ocean, (b) land, (c) Tropics (20°S- 20°N) and (d) Extratropics (40°S - 60°S and 40°N - 60°N).

*Competing interests.* The authors declare that they have no conflict of interest.

*Acknowledgements.* This study was supported by the European Research Council starting grant "QUAERERE" (GA 306284). The authors are grateful to the satellite retrieval science teams for the CALIPSO, CloudSat and MODIS instruments, and in particular Seiji Kato and his coworkers at NASA Langley for compiling the CCCM dataset and helping with questions. The authors wish to thank two anonymous reviewers for their constructive comments.

| Cloud Classes | | Low | | Middle | | High | |
|---|---|---|---|---|---|---|---|
| | | $N_\mathrm{d}$ | $L$ | $N_\mathrm{d}$ | $L$ | $N_\mathrm{d}$ | $L$ |
| Ocean | Homogeneous, mean | $67\,\mathrm{cm}^{-3}$ | $32\,\mathrm{g\,m}^{-2}$ | $43\,\mathrm{cm}^{-3}$ | $23\,\mathrm{g\,m}^{-2}$ | $55\,\mathrm{cm}^{-3}$ | $33\,\mathrm{g\,m}^{-2}$ |
| | median | $36\,\mathrm{cm}^{-3}$ | $18\,\mathrm{g\,m}^{-2}$ | $27\,\mathrm{cm}^{-3}$ | $14\,\mathrm{g\,m}^{-2}$ | $40\,\mathrm{cm}^{-3}$ | $20\,\mathrm{g\,m}^{-2}$ |
| | Inhomogeneous, mean | $46\,\mathrm{cm}^{-3}$ | $36\,\mathrm{g\,m}^{-2}$ | $28\,\mathrm{cm}^{-3}$ | $31\,\mathrm{g\,m}^{-2}$ | $40\,\mathrm{cm}^{-3}$ | $47\,\mathrm{g\,m}^{-2}$ |
| | median | $26\,\mathrm{cm}^{-3}$ | $21\,\mathrm{g\,m}^{-2}$ | $20\,\mathrm{cm}^{-3}$ | $18\,\mathrm{g\,m}^{-2}$ | $29\,\mathrm{cm}^{-3}$ | $27\,\mathrm{g\,m}^{-2}$ |
| Land | Homogeneous, mean | $163\,\mathrm{cm}^{-3}$ | $43\,\mathrm{g\,m}^{-2}$ | $141\,\mathrm{cm}^{-3}$ | $37\,\mathrm{g\,m}^{-2}$ | $174\,\mathrm{cm}^{-3}$ | $33\,\mathrm{g\,m}^{-2}$ |
| | median | $107\,\mathrm{cm}^{-3}$ | $24\,\mathrm{g\,m}^{-2}$ | $103\,\mathrm{cm}^{-3}$ | $23\,\mathrm{g\,m}^{-2}$ | $131\,\mathrm{cm}^{-3}$ | $23\,\mathrm{g\,m}^{-2}$ |
| | Inhomogeneous, mean | $145\,\mathrm{cm}^{-3}$ | $42\,\mathrm{g\,m}^{-2}$ | $121\,\mathrm{cm}^{-3}$ | $47\,\mathrm{g\,m}^{-2}$ | $129\,\mathrm{cm}^{-3}$ | $42\,\mathrm{g\,m}^{-2}$ |
| | median | $98\,\mathrm{cm}^{-3}$ | $26\,\mathrm{g\,m}^{-2}$ | $85\,\mathrm{cm}^{-3}$ | $32\,\mathrm{g\,m}^{-2}$ | $93\,\mathrm{cm}^{-3}$ | $30\,\mathrm{g\,m}^{-2}$ |
| Tropics | Homogeneous, mean | $42\,\mathrm{cm}^{-3}$ | $22\,\mathrm{g\,m}^{-2}$ | $35\,\mathrm{cm}^{-3}$ | $21\,\mathrm{g\,m}^{-2}$ | $87\,\mathrm{cm}^{-3}$ | $34\,\mathrm{g\,m}^{-2}$ |
| | median | $21\,\mathrm{cm}^{-3}$ | $13\,\mathrm{g\,m}^{-2}$ | $20\,\mathrm{cm}^{-3}$ | $12\,\mathrm{g\,m}^{-2}$ | $52\,\mathrm{cm}^{-3}$ | $22\,\mathrm{g\,m}^{-2}$ |
| | Inhomogeneous, mean | $33\,\mathrm{cm}^{-3}$ | $27\,\mathrm{g\,m}^{-2}$ | $28\,\mathrm{cm}^{-3}$ | $30\,\mathrm{g\,m}^{-2}$ | $66\,\mathrm{cm}^{-3}$ | $47\,\mathrm{g\,m}^{-2}$ |
| | median | $20\,\mathrm{cm}^{-3}$ | $15\,\mathrm{g\,m}^{-2}$ | $20\,\mathrm{cm}^{-3}$ | $18\,\mathrm{g\,m}^{-2}$ | $42\,\mathrm{cm}^{-3}$ | $30\,\mathrm{g\,m}^{-2}$ |
| Extratropics | Homogeneous, mean | $107\,\mathrm{cm}^{-3}$ | $42\,\mathrm{g\,m}^{-2}$ | $73\,\mathrm{cm}^{-3}$ | $30\,\mathrm{g\,m}^{-2}$ | $125\,\mathrm{cm}^{-3}$ | $31\,\mathrm{g\,m}^{-2}$ |
| | median | $50\,\mathrm{cm}^{-3}$ | $26\,\mathrm{g\,m}^{-2}$ | $45\,\mathrm{cm}^{-3}$ | $18\,\mathrm{g\,m}^{-2}$ | $89\,\mathrm{cm}^{-3}$ | $21\,\mathrm{g\,m}^{-2}$ |
| | Inhomogeneous, mean | $82\,\mathrm{cm}^{-3}$ | $48\,\mathrm{g\,m}^{-2}$ | $66\,\mathrm{cm}^{-3}$ | $42\,\mathrm{g\,m}^{-2}$ | $122\,\mathrm{cm}^{-3}$ | $38\,\mathrm{g\,m}^{-2}$ |
| | median | $45\,\mathrm{cm}^{-3}$ | $36\,\mathrm{g\,m}^{-2}$ | $37\,\mathrm{cm}^{-3}$ | $24\,\mathrm{g\,m}^{-2}$ | $92\,\mathrm{cm}^{-3}$ | $26\,\mathrm{g\,m}^{-2}$ |

**Table A1.** As Table 1, but separated for ocean, land, Tropics (20°S- 20°N) and Extratropics (40°S - 60°S and 40°N - 60°N).

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
