# Peer review of "A new classification of satellite derived liquid water cloud regimes at cloud scale"

_Atmospheric Chemistry and Physics, 2019_

## Referee Comment (RC1) · Anonymous Referee #1 · 15 Nov 2019

General comments:

This study proposes a new classification of liquid water clouds based on cloud-top height and cloud-base height, the former and latter being employed to quantify cloud heterogeneity and cloud altitude, respectively. The authors employ their newly developed retrieval for cloud-base height at cloud scale. The total six cloud categories are defined as a result from three cloud-base height intervals and two intervals of cloud-top horizontal variability. It is then shown that the climatology of their occurrence is reasonable on the global scale, and two basic cloud properties (liquid water path and cloud droplet number concentration) are also documented to show some interesting differences between the categories defined. I think that the authors' analysis is a meaningful addition to current knowledge of satellite-based analysis of cloud regimes,

particularly given that this study's approach of classification is based on cloud geometrical information and thus independent of cloud microphysical/optical properties. This will enable more meaningful investigation of cloud microphysical/optical properties for different cloud regimes as a function of environmental factors such as aerosol and stability conditions. I only have a couple of minor comments described below in an attempt to make the authors' analysis more sounding before the paper will be published in Atmos. Chem. Phys.

Specific comments:

Overall: It would be beneficial to readers if quantitative information about retrieval uncertainty of cloud-base height and cloud-top height is provided so that readers can evaluate how robust the statistics shown in the manuscript (e.g. RFOs of different categories and PDFs of cloud properties) are.

Line 165-170, Figure 4 and Table 1: The authors argue here that characteristics of Nd and L (in both PDFs and mean/median values) for six cloud categories show some signatures of different cloud behaviors between over continent and ocean. I would recommend the authors to separate the analysis into continent and ocean to more clearly see land-ocean differences in Nd and L in each category, facilitating the authors' interpretation and also eliminating the effect of background aerosol differences between land and ocean on cloud characteristics.

Technical comments:

Figure 1 right panel: The color bar appears to show relative scale of variability, but the text (Line 114) states that "Cloud top height variability is defined here as the average absolute deviation". Can you clarify?

Figure 2: Please add labels for horizontal axes, i.e. "cloud base height [m]" for left panel and "cloud-top height variability [%]" for right panel. It would also be helpful for readers if characters such as "low", "middle" and "high" are added to the corresponding ranges

[Figure]

of cloud-base height on left panel and those such as "stratiform" and "cumuliform" are added to the corresponding range of cloud-top height variability on right panel.

Line 153: "their response to perturbations": "perturbations" of what?
* * *

---

## Referee Comment (RC2) · Anonymous Referee #2 · 21 Nov 2019

The paper describes the use of CloudSat and CALIPSO retrievals of cloud base and cloud top heights to classify low (water) clouds in the atmosphere across the globe. The data are then combined with MODIS retrievals of number concentration and liquid water path to show the utility of the classification in distinguishing between different cloud states. The study is nice addition to the body of work on cloud classification from satellite observations. The paper is well-written and follows a logical flow of arguments. Its weaknesses lie in an incomplete description of the techniques applied as well as a lack of discussion of some obvious caveats in the results. I expand on both in my comments below. These issues should be straightforward to address and the paper will be acceptable for publication once they are.

Major comments

[Figure]

Description of the techniques used:

Lines 62-65: The paragraph introduces the CCCM data set that forms the foundation of this study. For the reader to fully comprehend how the classification works and how it might later link to the MODIS retrievals it is necessary to add a short paragraph here that provides a little more detail on how this merged product was created. How is the vertically resolved but horizontally sparse CloudSat and CALIPSO data combined with horizontally resolved MODIS data? What does it mean to use the CloudSat and CALIPSO data on a 20 km footprint? I realise there is a set of papers to read up on this, but not only would it help attract a greater readership if a short summary was provided here, it is also essential to understand the results that follow.

Line 117: You mention the use of adjacent footprints to look at the variability. What does adjacent mean here? Adjacent in which direction? I assume it is adjacent along the Aqua track? Please provide a little more detail here.

Discussion of caveats

The biggest caveat of the study is that the choice of looking at liquid water clouds only automatically introduces a geographic distribution to the result. This is so, because the 0C level in the tropics is at 5 (or so) km and in the polar regions it's very low (or even non-existent). This will bias liquid water path to be larger in the tropics and smaller in the extratropics. As the cloud classes show a strong geographic distribution (Figure 3), the separation of liquid water path by cloud class in Figure 4 could simply be a geographic sampling issue and have nothing to do with cloud-aerosol interactions. There is probably little you can really do about this at the definition stage, but you can account for it in the analysis of N_d and L and you MUST discuss that this as an issue in your conclusions.

I recommend to reduce this issue by performing analyses of the N_d and L pdfs in selected regions. As a minimum one might want to separate the (warm) tropics from the (cold) extra tropics. You could then contrast say land and ocean in both regions

and see if something useful emerges. That way you can alleviate at least some of the concern that the L differences are just difference in physical thickness resulting from the fact that different cloud classes preferably occur in the tropics (e.g., inhomogeneous C_B,mid) or the mid-latitudes (C_B,low).

Places where this fits in the manuscript are near line 165 or 175 and in the summary and conclusions.

Minor comments (on chronological order)

Line 29: Parentheses missing around the citations.

Line 96: Why do you use "roughly terciles" rather than terciles? How rough is rough? What are the tercile values compared to what you use? Presumably you just want round numbers or is there more to it?

Line 101: Add 'and' between homogeneous and cumuliform

Line 121: Actually, the shallow Cu regions show remarkably little variance compared to the deep tropics! As you are only using water clouds and therefore exclude deep convection, this is a little surprising. Is this a physical signal or is there something difficult about your method in the deep tropics that might cause this result? Please discuss this more deeply.

Line 124: The statement that your classification can separate stratiform and cumuliform is true to first order, but the trade regime, which is mostly cumuliform, does not show up very strongly. This makes sense as the inversions there are still quite strong and even though the clouds might be cumuliform, their tops are tightly constrained by the inversion height. So there is a caveat on your simple inhomogeneity assumption here, that you should acknowledge.

Line 130: Delete: 'Consistent with the expectation', as this certainly was not my expectation and might not be that of all readers.

Figure 3: Bottom right panel is mislabeled.

Line 136: The low H-base clouds nicely follow regions of low SST. It might be worth mentioning this here.

Line 139 and through the paper: I believe it is dangerous to refer to you classes as "mid-level clouds" or "high clouds", as they are not. Please be diligent in using terms like low-base, mid-level-base or high-base clouds. You do this in parts already and it is important to stick to that to minimise confusion with real mid-level or high clouds in the atmosphere. Please go carefully through the entire manuscript to fix this everywhere.

Summary and conclusions section: Please add a short discussion as what you perceive to be the weaknesses of your methodology and how those could be addressed in future work.

---

## Author Comment (AC1) · 17 Jan 2020

*We would like to thank the reviewer for the effort in helping us improve the manuscript. Below we respond point-by-point to the comments.*

General comments:
This study proposes a new classification of liquid water clouds based on cloud-top height and cloud-base height, the former and latter being employed to quantify cloud heterogeneity and cloud altitude, respectively. The authors employ their newly developed retrieval for cloud-base height at cloud scale. The total six cloud categories are defined as a result from three cloud-base height intervals and two intervals of

cloud-top horizontal variability. It is then shown that the climatology of their occurrence is reasonable on the global scale, and two basic cloud properties (liquid water path and cloud droplet number concentration) are also documented to show some interesting differences between the categories defined. I think that the authors' analysis is a meaningful addition to current knowledge of satellite-based analysis of cloud regimes, particularly given that this study's approach of classification is based on cloud geometrical information and thus independent of cloud microphysical/optical properties. This will enable more meaningful investigation of cloud microphysical/optical properties for different cloud regimes as a function of environmental factors such as aerosol and stability conditions. I only have a couple of minor comments described below in an attempt to make the authors' analysis more sounding before the paper will be published in Atmos. Chem. Phys.

*We thank the reviewer for this thoughtful and constructive statement.*

Specific comments:

Overall: It would be beneficial to readers if quantitative information about retrieval uncertainty of cloud-base height and cloud-top height is provided so that readers can evaluate how robust the statistics shown in the manuscript (e.g. RFOs of different categories and PDFs of cloud properties) are.

*This is a very good point and a very useful addition. Cloud base height was examined in detail by Mülmenstädt et al. (ESSD 2018), and we added a statement to the description of the data.*

Line 165-170, Figure 4 and Table 1: The authors argue here that characteristics of Nd and L (in both PDFs and mean/median values) for six cloud categories show some signatures of different cloud behaviors between over continent and ocean. I would recommend the authors to separate the analysis into continent and ocean to more clearly see land-ocean differences in Nd and L in each category, facilitating

the authors' interpretation and also eliminating the effect of background aerosol differences between land and ocean on cloud characteristics.

*We thank the reviewer for this very useful suggestion, and now show and discuss the characteristics separately for land and ocean in an Appendix, in conjunction with the discussion of Tropics vs. extratropics as suggested by reviewer 2.*

Technical comments:

Figure 1 right panel: The color bar appears to show relative scale of variability, but the text (Line 114) states that "Cloud top height variability is defined here as the average absolute deviation". Can you clarify?

*We thank the reviewer for this attentive reading of the text. Indeed, it is the relative deviation.*

Figure 2: Please add labels for horizontal axes, i.e. "cloud base height (m)" for left panel and "cloud-top height variability (%)" for right panel. It would also be helpful for readers if characters such as "low", "middle" and "high" are added to the corresponding ranges of cloud-base height on left panel and those such as "stratiform" and "cumuliform" are added to the corresponding range of cloud-top height variability on right panel.

*The figure is revised as suggested.*

Line 153: "their response to perturbations": "perturbations" of what?

*We thought specifically of aerosols and add this here.*

---

## Author Comment (AC2) · 17 Jan 2020

*We would like to thank the reviewer for the effort in helping us improve the manuscript. Below we respond point-by-point to the comments.*

The paper describes the use of CloudSat and CALIPSO retrievals of cloud base and cloud top heights to classify low (water) clouds in the atmosphere across the globe. The data are then combined with MODIS retrievals of number concentration and liquid water path to show the utility of the classification in distinguishing between different cloud states. The study is nice addition to the body of work on cloud classification from satellite observations. The paper is well-written and follows a logical flow of

[Figure]

arguments. Its weaknesses lie in an incomplete description of the techniques applied as well as a lack of discussion of some obvious caveats in the results. I expand on both in my comments below. These issues should be straightforward to address and the paper will be acceptable for publication once they are.
*We thank the reviewer for the careful evaluation of the manuscript.*

Major comments
Description of the techniques used:
Lines 62-65: The paragraph introduces the CCCM data set that forms the foundation of this study. For the reader to fully comprehend how the classification works and how it might later link to the MODIS retrievals it is necessary to add a short paragraph here that provides a little more detail on how this merged product was created. How is the vertically resolved but horizontally sparse CloudSat and CALIPSO data combined with horizontally resolved MODIS data? What does it mean to use the CloudSat and CALIPSO data on a 20 km footprint? I realise there is a set of papers to read up on this, but not only would it help attract a greater readership if a short summary was provided here, it is also essential to understand the results that follow.
*The reviewer is right that it is better to add a short statement about how the colleagues at NASA Langley did the work to create the CCCM dataset, rather than just referencing their papers. We added a paragraph explaining the details of the procedure.*

Line 117: You mention the use of adjacent footprints to look at the variability. What does adjacent mean here? Adjacent in which direction? I assume it is adjacent along the Aqua track? Please provide a little more detail here.
*The reviewer is correct with her/his assumption, and we now clarify this in the revised manuscript.*

Discussion of caveats

The biggest caveat of the study is that the choice of looking at liquid water clouds only automatically introduces a geographic distribution to the result. This is so, because the 0C level in the tropics is at 5 (or so) km and in the polar regions it's very low (or even non-existent). This will bias liquid water path to be larger in the tropics and smaller in the extratropics. As the cloud classes show a strong geographic distribution (Figure 3), the separation of liquid water path by cloud class in Figure 4 could simply be a geographic sampling issue and have nothing to do with cloud-aerosol interactions. There is probably little you can really do about this at the definition stage, but you can account for it in the analysis of N_d and L and you MUST discuss that this as an issue in your conclusions.

*The reviewer is right that this is an important caveat and we clearly now state it in the conclusions.*

I recommend to reduce this issue by performing analyses of the N_d and L pdfs in selected regions. As a minimum one might want to separate the (warm) tropics from the (cold) extra tropics. You could then contrast say land and ocean in both regions and see if something useful emerges. That way you can alleviate at least some of the concern that the L differences are just difference in physical thickness resulting from the fact that different cloud classes preferably occur in the tropics (e.g., inhomogeneous C_B,mid) or the mid-latitudes (C_B,low).

Places where this fits in the manuscript are near line 165 or 175 and in the summary and conclusions. *This is a very sensible suggestion. We now analyse and discuss separately Tropics and extratropics in a new Appendix, in conjunction with the discussion of oceanic vs. continental clouds as suggested by reviewer 1.*

Minor comments (on chronological order)
Line 29: Parentheses missing around the citations.
*Corrected, thanks!*

Line 96: Why do you use "roughly terciles" rather than terciles? How rough is rough? What are the tercile values compared to what you use? Presumably you just want round numbers or is there more to it?
*Indeed, this was just to use round numbers. We clarify it in the text.*

Line 101: Add 'and' between homogeneous and cumuliform
*This indeed needs better style. We chose to put a comma instead of an "and".*

Line 121: Actually, the shallow Cu regions show remarkably little variance compared to the deep tropics! As you are only using water clouds and therefore exclude deep convection, this is a little surprising. Is this a physical signal or is there something difficult about your method in the deep tropics that might cause this result? Please discuss this more deeply.
*Over the Pacific and Indian oceans, the variability in the shallow cu regions, in fact, is rather elevated; it is lower than in the Tropics mostly in the southern Atlantic ocean that we chose to center our figure on. We added a discussion, as recommended by the reviewer.*

Line 124: The statement that your classification can separate stratiform and cumuliform is true to first order, but the trade regime, which is mostly cumuliform, does not show up very strongly. This makes sense as the inversions there are still quite strong and even though the clouds might be cumuliform, their tops are tightly constrained by the inversion height. So there is a caveat on your simple inhomogeneity assumption here, that you should acknowledge.
*The reviewer is right, and we write a statement on this now in the revised manuscript.*

Line 130: Delete: 'Consistent with the expectation', as this certainly was not my expectation and might not be that of all readers.
*Done as suggested.*

Figure 3: Bottom right panel is mislabeled.
*The figure is revised as suggested.*

Line 136: The low H-base clouds nicely follow regions of low SST. It might be worth mentioning this here.
*Thanks for this, we do this in the revised manuscript.*

Line 139 and through the paper: I believe it is dangerous to refer to you classes as "mid-level clouds" or "high clouds", as they are not. Please be diligent in using terms like low-base, mid-level-base or high-base clouds. You do this in parts already and it is important to stick to that to minimise confusion with real mid-level or high clouds in the atmosphere. Please go carefully through the entire manuscript to fix this everywhere.
*The reviewer is right, and we follow this advice.*

Summary and conclusions section: Please add a short discussion as what you perceive to be the weaknesses of your methodology and how those could be addressed in future work.
*We take up the earlier comment by the reviewer and note this could be enhanced to characterize all clouds, not just the liquid ones. We also describe how the study might be enhanced by using passive retrievals of cloud-base height via triangulation from e.g. MISR.*

---

## Author Response (AR2)

We thank the Editor for carefully proofreading our revised manuscript and spotting the poor formulation. We corrected this in the new revision.

[revised manuscript text omitted]